# Oophorectomy in Premenopausal Patients with Estrogen Receptor-Positive Breast Cancer: New Insights into Long-Term Effects

Fatima Khan [1], Kristin Rojas [1,2], Matthew Schlumbrecht [1,3,4] and Patricia Jeudin [1,3,4,*]

1   Departments of Medicine, Leonard M. Miller School of Medicine, University of Miami, 1120 N.W. 14th Street, Miami, FL 33136, USA
2   Dewitt Daughtry Department of Surgery, University of Miami, Miami, FL 33136, USA
3   Sylvester Comprehensive Cancer Center, University of Miami, Miami, FL 33136, USA
4   Division of Gynecologic Oncology, University of Miami, Miami, FL 33136, USA
*   Correspondence: pjeudin@med.miami.edu

**Abstract:** Approximately 80% of breast cancers are estrogen receptor-positive (ER+), and 68–80% of those occur in premenopausal or perimenopausal women. Since the introduction of tamoxifen for adjuvant endocrine therapy in women with non-metastatic ER+ breast cancer, subsequent trials have demonstrated an oncologic benefit with the addition of ovarian function suppression (OFS) to adjuvant endocrine therapy. Subsequently, therapies to either suppress or ablate ovarian function may be included in the treatment plan for patients that remain premenopausal or perimenopausal after upfront or adjuvant chemotherapy and primary surgery. One strategy for OFS, bilateral salpingo-oophorectomy (BSO), has lasting implications, and the routine recommendation for this strategy warrants a critical analysis in this population. The following is a narrative review of the utility of ovarian suppression or ablation (through either bilateral oophorectomy or radiation) in the context of adjuvant endocrine therapy, including selective estrogen receptor modulators (SERMs) and aromatase inhibitors (AIs). The long-term sequelae of bilateral oophorectomy include cardiovascular and bone density morbidity along with sexual dysfunction, negatively impacting overall quality of life. As gynecologists are the providers consulted to perform bilateral oophorectomies in this population, careful consideration of each patient's oncologic prognosis, cardiovascular risk, and psychosocial factors should be included in the preoperative assessment to assist in shared decision-making and prevent the lifelong adverse effects that may result from overtreatment.

**Keywords:** bilateral salpingo-oophorectomy; breast cancer; ER+; tamoxifen; aromatase inhibitor; endocrine therapy; sexual dysfunction; vasomotor symptoms

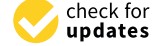



## 1. Introduction

Approximately eighty percent of breast cancers diagnosed in premenopausal women are estrogen receptor-positive (ER+) [1–3]. The estrogen receptor is a ligand-based transcription factor that is a key regulator of normal mammary development and differentiation. Estrogen receptors can be activated via estrogen-dependent or estrogen-independent mechanisms to promote the transcription of estrogen response elements (EREs) or transcription factor response elements (TRFEs). In ER+ breast cancer, cellular proliferation is driven by estrogen. Abnormal ER signaling is implicated in the majority of breast cancers, and is subsequently targeted by endocrine therapies (Figure 1). In addition to direct inhibition of ER+ breast cancers, the suppression of the body's overall production of estrogen by the ovaries in premenopausal women has been shown to improve survival [2]. The ovaries can be suppressed either with medication or through irreversible procedures. As such, there has been an increase in discourse in recent years regarding the status of ovarian suppression modalities in premenopausal women with ER+ breast cancer.

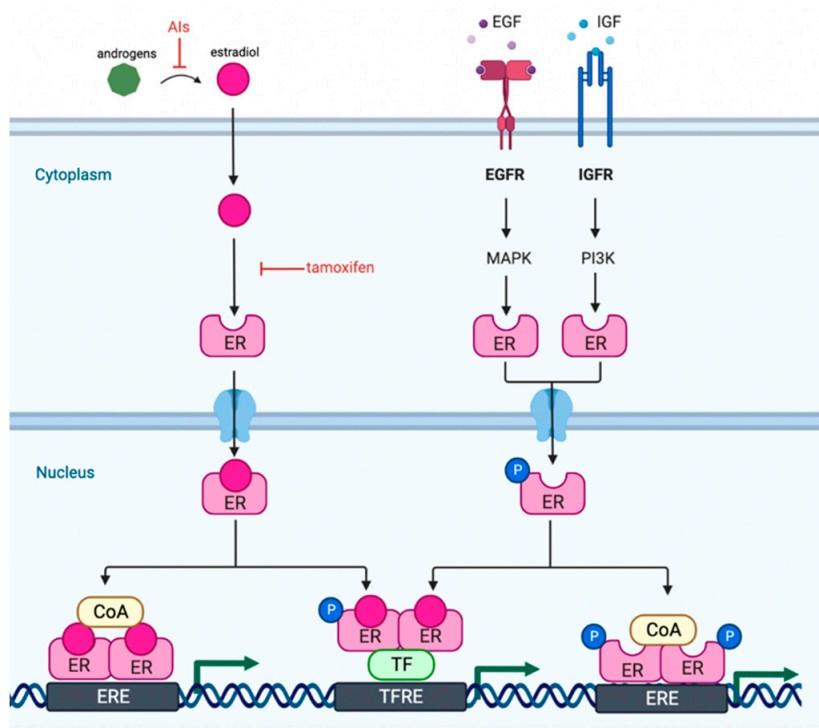

**Figure 1.** Estrogen receptor signaling. AI = aromatase inhibitor, EGF = epidermal growth factor, EGFR = epidermal growth factor receptor, IGF = insulin-like growth factor, IGFR = insulin-like growth factor receptor, ER = estrogen receptor, MAPK = mitogen-activated protein kinase, PI3K = phosphoinositide-3 kinase, ERE = estrogen response element, TF = transcription factor, TFRE = transcription factor response element.

Premenopausal breast cancer is associated with distinctive risk factors, tumor characteristics, and clinical outcomes that oftentimes warrant additional therapy [4]. There is a demonstrated benefit prophylactic bilateral salpingo-oophorectomy (BSO), or the removal of fallopian tubes and ovaries, for women with germline genetic mutations, such as BRCA1 or BRCA2. Some studies have also shown short-term hormone replacement therapy following BSO in some of these patients to be safe [5,6]. However, the risk–benefit ratio of BSO for premenopausal patients without hereditary breast and ovarian cancer (HBOC) syndrome warrants careful consideration due to the potential for long-lasting toxicities of acute hypoestrogenism occurring years before the natural onset of normal menopause.

A 2007 meta-analysis demonstrated a reduction in recurrence in women with ER+ breast cancer when an ovarian function suppressor (OFS) was combined with tamoxifen [7]. The addition of OFS to tamoxifen via a GnRH agonist, BSO, or ovarian radiation in the original analysis of the Suppression of Ovarian Function Trial (SOFT) did not result in an improvement in disease-free survival (DFS) [8]. A more recent analysis with a longer follow-up period found that the addition of OFS to tamoxifen improved the 8-year DFS from 78.9% to 93.2% (HR, 0.76; $p = 0.009$). However, women deemed "low-risk" and thus not requiring adjuvant chemotherapy in this trial still did extremely well on tamoxifen alone, with an overall survival of 98.8% [9].

The publication of this and subsequent trials demonstrating the benefit of OFS when added to adjuvant endocrine therapy has led to the addition of suppression or ablation of ovarian function for most women with premenopausal breast cancer. As such, medical oncologists may refer these patients to gynecologists for consideration of BSO, a single operation, instead of monthly or tri-monthly injections of a GnRH agonist to achieve suppression, especially in cases of continued menstruation after primary treatment [10].

BSO is associated with long-term adverse effects in both premenopausal and postmenopausal populations. These risks include cardiovascular disease, stroke, accelerated bone density loss, and mood disorders [11–14]. BSO prior to the age of 40 years is associated with a 48% increase in all-cause mortality (HR, 1.48; 95% CI, 1.03–2.13) [15]. The utilization of BSO as a component of breast cancer adjuvant therapy remains controversial. Although the 2021 National Comprehensive Cancer Network guidelines recommend five years of tamoxifen with or without OFS or AI with OFS/BSO as adjuvant endocrine therapy for patients with premenopausal ER+ breast cancer, the 2022 updated guidelines elaborate that the magnitude of benefit from BSO or radiation ablation in this population is similar to that achieved with cyclophosphamide–methotrexate–fluorouracil (CMF) therapy alone [16,17]. The risks and benefits of the long-term outcomes of BSO in premenopausal women with breast cancer warrants a thoughtful multidisciplinary discussion and shared decision-making.

Long-term survival continues to improve with the addition of adjuvant CDK 4/6 inhibitors to high-risk premenopausal patients with ER+ breast cancer; however, long-term sequelae of permanent ovarian ablation will vary from patient to patient [18]. As a result, cancer care providers must carefully weigh the risks and benefits of treatment components and the potential for long-term effects, particularly in younger women with an otherwise longer life expectancy. The following is a review to revisit the application and utility of BSO in the context of adjuvant endocrine suppression including selective estrogen receptor modulators (SERMs) and aromatase inhibitors (AIs), and to highlight the long-term sequelae of BSO among individuals with premenopausal ER+ breast cancer.

## 2. Methods

An electronic search was conducted to identify relevant publications in PubMed since the year 2000. The following (MESH) terms were used and combined: oophorectomy, salpingo-oophorectomy, breast cancer, breast neoplasms, endocrine therapy, premenopausal breast cancer, adverse effects, quality of life, cardiovascular effects, and bone density. Eligibility was assessed by FK and PJ, and discrepancies were resolved through discussion. Publications evaluating cancer-related outcomes of BSO after a diagnosis of ER+ premenopausal breast cancer and publications evaluating long-term effects of BSO regardless of cancer history were included. Specifically, we searched studies that provided a comparison of overall survival (OS) or disease-free survival (DFS) between individuals who received BSO and those in a control group. Regarding long-term effects, we focused on studies that provided adequate measurements of risk in individuals who received BSO compared to those in a control group. We also sought data examining cost and quality of life. Publications were considered for review if they were randomized controlled trials (RCTs), cohort studies, retrospective studies, reviews, and cost analyses. Publications were excluded if they: (1) were not in English; (2) evaluated BSO surgical technique only; (3) did not use BSO as the primary independent variable; (4) were case reports; (5) were nonclinical (i.e., basic science); or (6) included only women undergoing prophylactic surgery for HBOC syndrome.

## 3. Principal Findings

Our initial search yielded 446 publications, 417 of which were excluded after full-text screening, leaving 30 for inclusion in this review (Figure 2). The included publications fall into three main categories: utilizing OFS with selective estrogen receptor modulators (SERMs), utilizing OFS with aromatase inhibitors (AIs), and the general adverse effects associated with BSO (Table S1). According to the results of the SOFT/TEXT analysis, a direct comparison of tamoxifen therapy with or without OFS did not produce a significant difference in the recurrence rate.

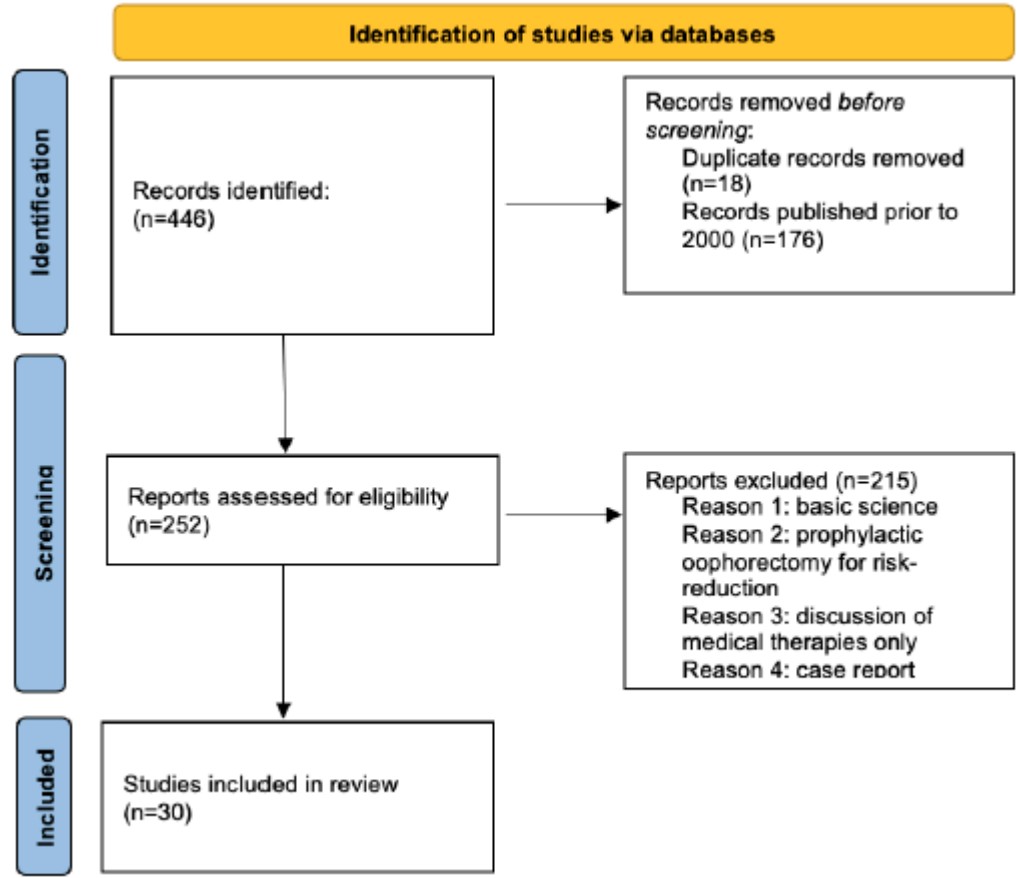

**Figure 2.** Overview of the studies to be included in the review.

Additional studies report similar or slightly improved outcomes with the addition of OFS to endocrine therapy, alongside high rates of adverse effects.

## 4. Results

### 4.1. Oophorectomy vs. OFS in Premenopausal Patients Receiving SERMs

Tamoxifen, a SERM, is an integral component of the adjuvant treatment of premenopausal ER+ breast cancer treatment, and is recommended by the NCCN Clinical Guidelines in Oncology and the 2014 American Society of Clinical Oncology (ASCO) Clinical Practice Guidelines based on level 1 evidence [19–21]. Several studies have examined the effects of OFS in conjunction with tamoxifen for the treatment of premenopausal ER+ breast cancer.

The Adjuvant Breast Cancer (ABC) Ovarian Ablation or Suppression Trial included 2144 participants with perimenopausal or premenopausal breast cancer, all of whom received tamoxifen with or without chemotherapy and who were randomized to receive either OFS with tamoxifen or tamoxifen alone. At 5.9 years, there was no significant difference in the OFS vs. non-OFS arms in relapse-free survival (HR, 0.95; 95% CI, 0.91–1.12; $p = 0.56$) or OS (HR, 0.94; 95% CI, 0.78–1.13; $p = 0.44$) [22].

In a sample of 709 premenopausal women with breast cancer, a 2008 phase III randomized trial found that seven-year DFS and OS were significantly improved after receipt of BSO plus tamoxifen compared to those of the control group ($p = 0.0003$ and $p = 0.0002$, respectively) [23]. Notably, the control group received observation only (no tamoxifen or adjuvant chemotherapy) despite approximately half of the individuals in this group having node-positive disease. Reliable conclusions about the benefit of BSO in this population are not possible due to the lack of a tamoxifen-alone comparison group, and the recurrence rates do not reflect those of patients treated with contemporary therapies.

The 2014 phase III Eastern Cooperative Oncology Group (ECOG 3193) trial randomized 345 women with node-negative ER+ breast cancer to either tamoxifen alone or tamoxifen and OFS, where none of the enrolled patients received chemotherapy. There was no significant difference in the 5-year rates of DFS or OS between the arms. Notably, grade 3 toxicities were twice as high in patients who received OFS (22.4% vs. 12.3%, $p = 0.004$) [24].

The combined analysis of the randomized Tamoxifen and Exemestane Trial (TEXT) and the Suppression of Ovarian Function Trial (SOFT) sought to determine the benefit of the addition of OFS to either tamoxifen or exemestane, an aromatase inhibitor, in premenopausal women with early-stage ER+ breast cancer. The 8-year results from SOFT found a disease-free survival (DFS) of 78.9% with tamoxifen alone, 83.2% with tamoxifen plus OFS, and 85.9% with exemestane plus OFS ($p = 0.009$) [9]. However, the DFS of tamoxifen plus OFS was not significantly different than that of tamoxifen alone (HR, 0.76; 95% CI, 0.52–1.12). Roughly half of these recurrences in the premenopausal women were distant recurrences. There was also a higher risk of adverse events in the two OFS treatment arms, 31.0% and 32.3%, compared to a rate of 24.6% in the control (tamoxifen alone) arm.

The 2020 ASTRRA trial was a randomized phase III study allocating 1483 premenopausal women with ER+ breast cancer to either five years of tamoxifen alone or five years of tamoxifen with OFS for two years. Most patients received OFS through a monthly injection of a gonadotropin-releasing hormone (GnRH) agonist. The five-year DFS was significantly greater in the group that received an additional two years of OFS (91.1% vs. 87.5%; HR, 0.69; 95% CI, 0.48–0.97; $p = 0.033$) [25]. However, premenopausal women over the age of 45 years were excluded from the study. In a follow-up of the study, the estimated eight-year DFS rate was 85.4% in the tamoxifen with OFS group compared to 80.2% in the tamoxifen-only group (HR, 0.67; 95% CI, 0.51 to 0.87) [26]. There were no significant differences in OS between the two groups after eight years.

Overall, the addition of OFS confers a greater DFS and OS in many of these studies. While these studies did not directly compare BSO to non-surgical methods of OFS, the statistically significant differences in DFS in these studies were achieved largely by non-surgical methods of OFS. Therefore, the irreversibility of BSO and its associated risks may not afford an additional oncologic benefit, especially in high-risk women who are likely to receive chemotherapy or CDK 4/6 inhibitors in the neoadjuvant or adjuvant setting [18,27].

Many scholarly reviews also offer perspectives regarding the value added by OFS. In a review of OFS combined with various adjuvant endocrine therapy modalities, Jain et al. determined that the benefits of OFS plus tamoxifen are best achieved among higher-risk populations, defined as women under 40 years with higher grade and stage, lymph-node-positive disease, among other factors [28]. Citing the ABCSG-12 and TEXT/SOFT trials, the review concludes that advances in DFS provided by adding OFS to endocrine therapy are attenuated in average-risk populations, and are not without long-term adverse effects.

While OFS shows gains in DFS when added to tamoxifen in younger patients with more advanced disease who may not be candidates for chemotherapy, the benefits of surgical OFS in average-risk women in resource-rich settings are less likely, and need to be evaluated on a case-by-case basis.

### 4.2. Oophorectomy vs. OFS in Premenopausal Patients Receiving AIs

Aromatase inhibitors (AIs) have become a standard of care for the adjuvant endocrine treatment of most postmenopausal women with early-stage ER+ breast cancer. Although the use of aromatase inhibitors alone is generally contraindicated in premenopausal women due to the stimulation of estrogen production in functioning ovaries, the combination of OFS with an AI in premenopausal women is a potential strategy for adjuvant therapy for high-risk women.

There are very few studies that directly compare OFS and ovarian ablation through surgery or radiation when combined with an aromatase inhibitor. In the landmark MONARCH-E trial, high-risk patients with ER+/Her2- breast cancer were randomized to receive either adjuvant endocrine therapy alone or adjuvant endocrine therapy with

the CDK 4/6 inhibitor abemaciclib. "High-risk" was defined as having four or more positive axillary lymph nodes on surgical pathology or one to three positive nodes with any of the following: tumor size ≥ 5 cm, grade 3, or high ki-67 (>20%). Endocrine therapy included AI+/-OFS and tamoxifen +/-OFS, depending on the patient's menopausal status. Approximately 45% of the patients were premenopausal, but only half received OFS. The addition of the CDK 4/6 inhibitor abemaciclib to endocrine therapy resulted in a 25% reduction in the risk of an invasive event over a median follow-up period of 15.5 months. This risk reduction was even greater (37%) in patients who were premenopausal at the time of randomization. Comparative DFS analyses between therapy strategies, such as tamoxifen alone vs. endocrine therapy with OFS, in the premenopausal subgroup have not been reported.

However, any benefit to one strategy over another is tempered by the benefit afforded with the addition of the CDK 4/6 inhibitor [18]. As such, it is unlikely that BSO would add a significant treatment benefit to endocrine therapy in the context of neoadjuvant chemotherapy and adjuvant CDK 4/6 inhibition.

In a retrospective analysis of 66 premenopausal women with recurrent or metastatic breast cancer, BSO plus AI treatment did not result in a significantly higher progression-free survival when compared to patients who received OFS with a GnRH antagonist plus an AI (17.2 months vs. 13.3 months, $p = 0.245$). Most patients in both arms achieved a clinical response after six months, and there was a significantly higher proportion of patients in the BSO group compared to the GnRH group (88% vs. 69%, $p = 0.09$) [29]. Interestingly, the investigators found that having a higher ki67 (≥14%) or a luminal B-like feature were independent prognostic factors for a shorter PFS. Given the limitations of the data, BSO does not appear to be superior to ovarian function suppression with a GnRH antagonist in premenopausal patients with recurrent or incurable disease.

### 4.3. Adverse Effects of Early Oophorectomy

#### 4.3.1. Cardiovascular Effects

Several studies have noted an association between early BSO and an increased risk of cardiovascular death, which is the leading cause of death in women living in the United States [30–32]. In a population-based cohort study of over 144,000 postmenopausal women, the risk of cardiovascular disease (CVD) at seven years was greatest in women with surgically induced premature menopause (HR, 1.87), followed by those who underwent natural premature menopause (6.0%; HR, 1.36), in comparison to those who underwent menopause between the ages of 45 and 55 (3.9%) ($p < 0.001$) [33,34]. Though there has been mixed evidence in support of BSO as a definitive risk factor for CVD, a recent study published by the Mayo Clinic found a statistically significant association between BSO before the age of 45 years and cardiovascular death (HR, 1.44; 95% CI, 1.01–2.05), with an even higher risk in women who were not treated with estrogen until at least the age of 45 [19,35,36]. This increased risk of CVD from BSO may be further modified by a family history of myocardial infarction. A recent retrospective analysis found that cardiovascular mortality was greater among women with both BSO prior to the age of 45 years and a family history of myocardial infarction compared to those without either predisposing factor (HR, 2.88; 95% CI, 1.72–4.82) [37]. A review of the cardiovascular risks of each patient must be considered when counseling women with premenopausal breast cancer for whom BSO is recommended as a method of OFS, as well as younger women at increased risk of ovarian cancer who opt for risk-reducing salpingo-oophorectomy in the context of a pathogenic mutation.

#### 4.3.2. Bone Density

Given that estrogens and androgens inhibit bone resorption, the decline in estrogen from natural menopause or BSO has significant implications for bone health. A phase II randomized controlled trial of 80 women with breast cancer not on aromatase inhibitors but with a history of BSO demonstrated a mean reduction in bone mineral density of 8.5%

in the lumbar spine and 5.7% in the hips over 18 months, which translates to an increased risk of fracture [38].

Additionally, a prospective study of 270 Asian women previously treated with BSO for premenopausal breast cancer found a reduction in bone mineral density of 4.7% in the lumbar spine [39]. Taken together, these results support the increased fracture risk associated with premenopausal women who undergo BSO. This is still an active area of research, as evidenced by the Study of Bone Mineral Density in Women with Breast Cancer Treated with Triptorelin and Tamoxifen or Exemestane on Protocol IBCSG 25-02.

### 4.3.3. Quality of Life

The effects of BSO have the potential to impact overall quality of life in several aspects, including vasomotor symptoms. BSO is an independent predictor of vasomotor symptoms in postmenopausal women treated for breast cancer (aOR, 1.77; 95% CI, 1.37–2.27), and was the most prominent quality-of-life symptom reported among women under the age of 35 years in the TEXT/SOFT trials [40]. In a cohort of 896 women diagnosed with breast cancer at age 40 or younger, ER-positive disease treated with BSO or medical OFS was independently associated with a symptomatic, rather than an asymptomatic, trajectory of sexual functioning over the first 5 years following breast cancer diagnosis ($p < 0.05$, two-sided) [41]. BSO, as a permanent strategy for OFS, impacts overall quality of life through vasomotor, psychological, and sexual mechanisms. In those who experience symptoms, treatment-noncompliance rates can be high, highlighting the importance of addressing these sequelae. As systemic hormone replacement therapy after surgical menopause is usually contraindicated in those with a history of breast cancer, the permanence of BSO vs. temporary OFS should be weighed carefully in very young patients. While severely symptomatic patients on OFS may be allowed a "drug holiday" from their ovarian suppression or a change in therapy with improvement in their symptoms, this option is not available for patients who are surgically menopausal.

### 4.3.4. Overall Mortality

In the Nurses' Health Study, an ongoing prospective cohort study of over 15,000 women, BSO at any age for any indication was associated with an increased risk of death from all causes (HR, 1.13; 95% CI, 1.06–1.21) [36]. This elevated risk of death from all causes persisted in women who underwent BSO for any indication over the age of 60 years (HR, 1.31; 95% CI, 0.98–1.75). The same trend exists in other studies, where BSO is associated with an increased all-cause mortality in women undergoing BSO prior to the age of 45 years [42–44]. The risk of recurrence should be weighed against the increased overall mortality. As a comparison, the risk of mortality from any cancer in women aged 35–44 years is 19.6%, of which breast cancer comprises 7.2%. For comparison, mortality from cardiovascular disease accounts for 15.9% of deaths in this age group [45,46]. Although these studies do not analyze subgroups of women who underwent BSO as part of treatment for premenopausal breast cancer, there is evidence to suggest that oophorectomy at any age confers an increased risk of all-cause mortality.

## 5. Research Implications

While our analysis illustrates either modest or no improvement in survival associated with OFS in patients with ER+ premenopausal breast cancer, additional retrospective analyses are necessary to elucidate the prognosis and sequelae associated with BSO exclusively. There is an ongoing clinical trial (NCT02440230) that aims to assess the safety of OFS with anastrozole compared to that of OFS with exemestane in Chinese premenopausal breast cancer patients. However, there is no treatment arm that does not receive OFS, and like in SOFT/TEXT trials, the method of OFS includes a mix of goserelin, a LHRH agonist, ovarian irradiation, and BSO.

## 6. Clinical Implications

A 2016 review by Love et al. concluded that BSO with tamoxifen is an effective therapy for ER+ premenopausal breast cancer, particularly in resource-limited settings, as one-third of annual new cases of breast cancer globally are now hormone receptor-positive tumors in premenopausal women from low- and middle-income countries [47]. As such, accounting for the social determinants of health to provide equitable care should factor into the treatment decision.

Additionally, BSO is associated with an increased risk of cardiovascular death, especially amongst women with a family history of CVD. Furthermore, BSO increases the risk of osteoporotic fracture and impacts sexual function. Thus, the role of BSO for adjuvant permanent ovarian ablation in premenopausal women with ER+ breast cancer, considering the long-term sequelae, should involve a thoughtful multidisciplinary discussion between providers along with patient-shared decision-making.

The value added to endocrine therapy by BSO has also been examined under an economic lens. In a cost–benefit simulation model, tamoxifen alone was associated with the highest life expectancy gain (18 years) at the lowest cost (USD 1566) [48]. Adding BSO led to a life expectancy gain of only three months in contrast with tamoxifen alone in women with prior chemotherapy for breast cancer [49]. Another cost-effectiveness analysis found a lower total mean cost per patient associated with BSO compared to a GnRH agonist [50]. Additionally, BSO is thought to reduce the burden of treatment, particularly in younger women, by eliminating the need for routine injections. However, the permanence of BSO cannot be underemphasized.

## 7. Limitations

The challenge in determining the isolated potential benefit from BSO as an OFS strategy is that in large landmark trials that provide rich data, BSO and medical OFS are often analyzed together, making it difficult to ascertain differences between these subsets of OFS. Furthermore, only a handful of studies have directly examined ovarian suppression or ablation against BSO in the context of breast neoplasms, as both methods have been shown to be effective in suppressing ovarian function [51,52]. Therefore, the conclusions reached in this review assume that OFS is equivalent to BSO in efficacy and long-term effects, with the difference namely being the irreversibility of BSO.

A recurring theme in many of the studies included in this review is the conclusion that BSO is best reserved for individuals with high-risk disease. However, "high-risk" can be a subjective determination based on the variety of different criteria used in these studies, including node status, grade and stage, and the age of the patient. Additionally, the conclusions made in the landmark trials we have discussed are based on relatively small samples of participants. In SOFT, only 8% of the patients undergoing OFS received BSO, making the results of this trial difficult to generalize when considering BSO specifically.

## 8. Conclusions

Premenopausal women diagnosed with ER+ breast cancer represent a cohort with unique challenges apart from coping with the stress of a diagnosis. Long-term survival in women living with breast cancer is steadily increasing due to improvements in early diagnosis and advances in cancer treatments and surveillance [52]. Despite being a higher-risk population due to presentation at a more advanced stage and higher grade, premenopausal patients may be particularly vulnerable to quality-of-life-impacting side effects, which may impact treatment compliance. BSO is therefore thought to be particularly effective in reducing the risk of recurrence in women aged between 35 and 40 years or in whom chemotherapy is recommended. However, with recent advances in targeted therapies for women with high-risk ER+ breast cancer, the benefit of OFS may be less significant in more modern cohorts, as reflected by the updated NCCN guidelines. These nuances highlight the burden of treatment decision-making tasked to medical oncologists, who must balance

the potential for improved DFS with adverse effects of ovarian function suppression and the risk of treatment noncompliance with induced menopause.

**Supplementary Materials:** The following supporting information can be downloaded at: https://www.mdpi.com/article/10.3390/curroncol30020139/s1, Table S1: BSO vs. OFS in Premenopausal Patients Receiving SERMs.

**Author Contributions:** F.K. contributed to the literature review, content, and design of the manuscript. K.R., M.S., and P.J. contributed to the conceptualization, interpretation, editing, final drafting, and clinical relevance of the manuscript. All authors have read and agreed to the published version of the manuscript.

**Funding:** This research received no external funding.

**Conflicts of Interest:** The authors declare no relevant conflict of interest.

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
