# Peer review of "Oophorectomy in Premenopausal Patients with Estrogen Receptor-Positive Breast Cancer: New Insights into Long-Term Effects"

_curroncol, doi:10.3390/curroncol30020139_

Round 1

Reviewer 1 Report

Dear Academic Editor,

thank you very much for the possibility to review this interesting manuscript about the utility of ovarian function suppression in a cohort of patients with Estrogen-Receptor positive breast cancer. I think that the results of this narrative review are most important for the knowledge of the topic. Furthermore, I think that this review is well written and understable. 
however, I have some suggestions for the authors:

1. Please, it is necessary to follow PRISMA guidelines.
2. Add a paragraph with the limitations of the review.

3. check the manuscript for typos and English language.

Author Response

    1. Please, it is necessary to follow PRISMA guidelines.

    We have included a PRISMA diagram in the body of the manuscript. This was intended to serve as a narrative rather than systematic review, and data was not aggregated or analyzed as part of a meta-analysis. However, we have provided additional information about our database search to further clarify our study selection methodology. Items 2-10 on the 2020 PRISMA checklist have been included.

    1. Add a paragraph with the limitations of the review.

    We agree with the necessity of discussing the limitations of this review. We have added more information to our existing limitations section (lines 326-336).

    1. check the manuscript for typos and English language.

    We have checked for typos and grammatical errors and updated accordingly.

Thank you so much for your time and feedback. We look forward to your responses.

Reviewer 2 Report

The article has an important topic for women: breast cancer

The article must suffer a few changes

1. please deeply detail the relationship between breast cancer and ER+

2. Also a scheme or a figure regarding breast cancer and ER+ will have a great impact for readers

3. please draw a table with all clinical studies included in the review

Author Response

  1. deeply detail the relationship between breast cancer and ER+

We have included in more detail the role of ER in breast cancer in the introduction.

  1. Also a scheme or a figure regarding breast cancer and ER+ will have a great impact for readers

We have included a basic scheme of estrogen receptor signaling and the role of tamoxifen and aromatase inhibitors.

  1. please draw a table with all clinical studies included in the review

We plan to include a table with the included studies and which category they fall into.

Round 2

Reviewer 1 Report

Dearest Academic Editor,

thank you for the opportunity to review this interesting manuscript. I believe that the authors have made a good effort to improve this work and, in addition, I believe it would be even more improved if references about BRCA1/2 mutated patients were added. Below I list the most recent papers indicated: 

1) Loizzi V, Dellino M, Cerbone M, Arezzo F, Chiariello G, Lepera A, Cazzato G, Cascardi E, Damiani GR, Cicinelli E, Cormio G. Hormone replacement therapy in BRCA mutation carriers: how shall we do no harm? Hormones (Athens). 2023 Jan 13. doi: 10.1007/s42000-022-00427-1. Epub ahead of print. PMID: 36637775.

2) Oh M, McBride A, Bhattacharjee S, Slack M, Jeter J, Abraham I. Economic value of knowing BRCA status: BRCA testing for prostate cancer prevention and optimal treatment. Expert Rev Pharmacoecon Outcomes Res. 2023 Jan 17. doi: 10.1080/14737167.2023.2169137. Epub ahead of print. PMID: 36649640.

3) Loizzi V, Dellino M, Cerbone M, Arezzo F, Cazzato G, Damiani GR, Pinto V, Silvestris E, Kardhashi A, Cicinelli E, Cascardi E, Cormio G. The Role of Hormonal Replacement Therapy in BRCA Mutated Patients: Lights and Shadows. Int J Mol Sci. 2023 Jan 1;24(1):764. doi: 10.3390/ijms24010764. PMID: 36614207; PMCID: PMC9821191.

4) Jang J, Kim Y, Kim JH, Cho SM, Lee KA. Cost-Effectiveness Analysis of Germline and Somatic BRCA Testing in Patients With Advanced Ovarian Cancer. Ann Lab Med. 2023 Jan 1;43(1):73-81. doi: 10.3343/alm.2023.43.1.73. Epub 2022 Sep 1. PMID: 36045059; PMCID: PMC9467835.